# Heterotrimeric G Protein-Mediated Signaling Is Involved in Stress-Mediated Growth Inhibition in *Arabidopsis thaliana*

**DOI:** 10.3390/ijms241311027

**Published:** 2023-07-03

**Authors:** Soeun Yang, Seohee Jung, Horim Lee

**Affiliations:** Department of Biotechnology, Duksung Women’s University, Seoul 03169, Republic of Korea; dosl68997719@duksung.ac.kr (S.Y.); sujjang0412@duksung.ac.kr (S.J.)

**Keywords:** G protein signaling, stress-mediated growth, trade-off

## Abstract

Heterotrimeric G protein-mediated signaling plays a vital role in physiological and developmental processes in eukaryotes. On the other hand, because of the absence of a G protein-coupled receptor and self-activating mechanism of the Gα subunit, plants appear to have different regulatory mechanisms, which remain to be elucidated, compared to canonical G protein signaling established in animals. Here we report that *Arabidopsis* heterotrimeric G protein subunits, such as Gα (*GPA1*) and Gβ (*AGB1*), regulate plant growth under stress conditions through the analysis of heterotrimeric G protein mutants. Flg22-mediated growth inhibition in wild-type roots was found to be caused by a defect in the elongation zone, which was partially blocked in *agb1-2* but not *gpa1-4*. These results suggest that *AGB1* may negatively regulate plant growth under biotic stress conditions. In addition, *GPA1* and *AGB1* exhibited genetically opposite effects on *FCA*-mediated growth inhibition under heat stress conditions. Therefore, these results suggest that plant G protein signaling is probably related to stress-mediated growth regulation for developmental plasticity in response to biotic and abiotic stress conditions.

## 1. Introduction

Heterotrimeric G proteins consisting of Gα, Gβ, and Gγ subunits regulate various signaling pathways involved in a wide range of physiological processes in eukaryotes [1,2]. In animals, several types of Gα, Gβ, and Gγ can form a variety of heterotrimeric G protein complexes with a seven-transmembrane (7TM) G protein-coupled receptors (GPCRs), which are the largest family of proteins in humans and recognize the external signals, including nutrients, neurotransmitters, and hormonal growth factors, for cellular responses [2,3]. The *Arabidopsis* genome contains one canonical Gα (*GPA1*), three extra-large G proteins (XLG1, 2, and 3), one Gβ (*AGB1*), and three Gγ, such as *AGG1*, *AGG2*, and *AGG3* [1,4,5,6,7,8], indicating lower complexity of the heterotrimeric G protein complexes in plants than in animals. Moreover, G protein signaling in plants may be simpler or regulated differently compared to animals.

Although plant and animal G protein signaling modules have similar structures through the same composition in complex, the behavior of heterotrimeric G protein subunits in plants may be different from signaling processes established in animals. The most striking difference in plants is the loss of the canonical 7TM GPCR coupled with the heterotrimeric G protein complex [9]. In animals, when signals, such as light, hormones, and ions, bind to the GPCR, the GDP-bound Gα subunit in the heterotrimeric G protein complex exchanges to the GTP-bound Gα subunit. The G protein complex is then dissociated into an active Gα and Gβγ dimer to activate effectors, such as adenylyl cyclase (AC), phospholipase C (PLC), and ion channel [2,3]. Hence, ligand-bound GPCRs may act as nucleotide exchange factors controlling the dissociation of active Gα and Gβγ subunits from inactive heterotrimeric G protein complexes in animals. However, plant heterotrimeric G proteins are usually the GPCR-independent mechanism and have a spontaneous self-activating property because of the intrinsically rapid nucleotide exchange from GDP to GTP and the slow GTP-hydrolytic activity of the Gα subunit [1,10]. In addition, instead of the canonical GPCR, plants have a 7TM regulator of G protein signaling-1 (RGS1), which encodes GTPase-accelerating proteins (GAP), stimulating GTPase activity of Gα for terminating the signaling event [1,10]. Therefore, signal transduction in plants is initiated by the self-activation of the Gα subunit GPA1, and the Gα and Gβγ subunits are dissociated to activate downstream effectors. A previous study reported that _D_-glucose acts as a candidate ligand for RSG1 to inhibit the GAP activity through the dissociation of RGS1 and GPA1 [11,12], suggesting that the deactivation of spontaneously activated G-proteins through RGS1 is prevented by the presence of a ligand in plants [1,9]. These results showed that G protein signaling in plants has a different regulatory system and downstream effectors, which are still poorly understood.

Despite the relatively simple composition of heterotrimeric G protein subunits in plants, the biological functions of G protein signaling are involved in a broad range of plant developmental growth, including seed germination, lateral root, hypocotyl elongation, and leaf expansion, as well as biotic/abiotic stress responses based on the analysis of G protein mutants, such as *gpa1* and *agb1* [10,13]. For example, the *gpa1-4* mutants showed a significantly lower germination rate under osmotic stress than wild-type (WT) [14]. In contrast, they exhibited higher fresh weight, chlorophyll contents, and total root lengths than WT under low-nitrogen conditions through direct interactions with the nitrate transporter (AtNRT1.4) and autophagy-related protein (AtATG8a) [15]. These findings suggest that the Gα subunit *GPA1* is positively involved in osmotic stress resistance and negatively regulates low nitrogen stress conditions. The *agb1-2* mutants exhibited rapid senescence of seedlings and bleached chlorophylls compared to WT under salt stress conditions [16], while they had a higher survival rate than WT under drought stress conditions [17], indicating that the Gβ subunit *AGB1* plays a promoted and negative role in regulating the salt and drought stress conditions, respectively. Interestingly, these results show that the same G protein, such as *GPA1* and *AGB1*, can regulate downstream processes positively or negatively. Although an analysis of the G protein mutants showed complex phenotypes depending on the development and stress conditions, all those responses might be correlated with developmental processes under normal growth conditions or stress tolerance related to growth. These suggest that G protein signaling may modulate plastic plant growth and development against everchanging biotic/abiotic environmental conditions. 

In this study, we show that the biological roles of heterotrimeric G protein subunits, such as *GPA1* and *AGB1*, are involved in the regulation of plant growth under stress conditions. By analyzing G protein mutants under stress conditions, we show that the *agb1-2* and *gpa1-4 agb1-2* mutants are partly resistant to flg22-mediated stress conditions, whereas *gpa1-4* and *gpa1-4 agb1-2* mutants respond to the *FCA*-mediated growth regulation under heat stress conditions. In addition, the effects of flg22-mediated growth inhibition in *agb1-2* and *FCA*-mediated growth inhibition in *gpa1-4* were found at the elongation zone and the division zone in roots, respectively. These results suggest that the regulatory mechanisms of G protein signaling have different effects under biotic and abiotic stress conditions for balanced plant growth and development.

## 2. Results

### 2.1. The agb1-2 Mutant Shows a More Resistant Root Growth in Response to flg22 Than WT 

The biological function of G protein signaling involved in growth retardation related to pathogen stresses was investigated by growing the WT Col-0, and *gpa1-4 agb1-2*, *agb1-2* and *gpa1-4* mutant seedlings on a half MS plate without (control) or with 100 nM flg22, respectively. The WT seedlings showed a significant decrease in root growth in response to the flg22 treatment (Figure 1). Interestingly, *gpa1-4* mutant seedlings displayed a similar reduction of root growth compared to that of WT (Figure 1), whereas the root growth of *agb1-2* single and *gpa1-4 agb1-2* double mutant seedlings was partially insensitive to the flg22 treatment (Figure 1). These results suggest that the growth retardation by pathogen stresses is partially dependent on the role of the G-protein signaling component AGB1. 

In *Arabidopsis*, seedling growth inhibition caused by microbe-associated molecular patterns (MAMPs), such as flg22 and elf26, is known as a late response with callose deposition [18,19]. After perception of MAMPs by pattern-recognition receptors (PRRs), such as FLAGELLIN SENSITIVE2 (FLS2), a very early response, including ion fluxes, oxidative burst, and mitogen-activated protein kinase (MAPK) activation, is induced, followed by ethylene biosynthesis, PRR endocytosis, and downstream gene activation as an early response within 30 min [19]. As shown in Figure 2, a very early response, such as MAPK activation, was triggered by the treatment of flg22 signals (Figure 2A), and early responsive downstream genes, such as *WRKY30* and *MYB51*, were also activated in WT (Figure 2B). Because the expression of downstream genes was not induced by a flg22 treatment in *fls2* mutant seedlings (Figure 2B), these results indicate that these responses occur rapidly after PRR FLS2 activation by flg22. In contrast to growth inhibition in WT and G protein mutants under an flg22 treatment, very early/early responses, including MAPK activation and gene expression, were indistinguishable between WT and single/double G protein mutants (Figure 2). Therefore, these results suggest that the simultaneous occurrence of MAMPs-triggered late responses is less tightly involved in the flg22 response pathway than the very early/early responses. In addition, this suggests that different signaling branches via heterotrimeric G proteins are required for seedling growth inhibition by flg22 in the pathogen response pathway.

### 2.2. The agb1-2 Mutant Shows a Less Sensitive Defect in the Elongation Zone in Response to flg22 

To dissect flg22-mediated seedling growth inhibition shown in roots, we observed the physiological and morphological effects on root growth and development. Interestingly, the change in the division zone indicated by the 5-ethynyl-2ʹ-deoxyuridine (EdU)-labeled dividing cells was unaffected by the treatment of flg22 compared to control in root meristems between WT and other G protein mutants (Figure 3A,B). Although the length of the division zone was reduced slightly by the treatment of 100 nM flg22 but not 10 nM flg22, those were not statistically significant (Figure 3B). In addition, the distal differentiation generated from columella stem cells (CSCs) was also unaffected by the flg22 treatment (Figure 3C). By Lugol’s staining analysis, WT and G protein mutants showed similar accumulation of starch granules at four or five layers, including one layer of differentiating CSC daughters and three or four layers of fully differentiated columella root cap cells [20]. These results suggest that flg22-mediated root growth inhibition is not caused via stem cell homeostasis, indicated by the division rate in root meristems and differentiation in root caps. 

Next, the elongation zone on root growth was observed by measuring the distance between the root cap and the differentiated root hair growing at least 50 µm length to investigate the cause of root growth inhibition. As shown in Figure 4A, the length until the elongation zone, including the division zone and transition zone, was reduced dramatically by the flg22 treatment in WT. Because the role of the division zone was eliminated as an estimated cause of root growth inhibition by flg22 (Figure 3A,B), these results suggest that flg22-mediated root growth inhibition is probably due to the reduction of the elongation zone. Similarly, the reduced elongation zone in roots was also observed clearly in *gpa1-4*, but not *agb1-2* and *gpa1-4 agb1-2* (Figure 4A). Although the reduction of elongation zone in *agb1-2* was statistically significant, the decreased rate of elongation zone in *agb1-2* and *gpa1-4 agb1-2*, showing 26% and 14%, respectively, under flg22-treated conditions compared to each control condition, was less sensitive than WT and *gpa1-4*, showing 46% and 41% of the decreased rate, respectively (Figure 4B). Since the primary root length of *agb1-2* and *gpa1-4 agb1-2* was also less sensitive than WT and *gpa1-4* (Figure 1B), these results demonstrate that flg22-mediated root growth inhibition may be caused by the lesion in the elongation zone.

### 2.3. The Response of PIF4 Expression in the agb1-2 Mutant Is Less Sensitive to flg22

The rapid expansion of the elongation zone contributes to primary root growth through the regulation of plant hormones, including auxin [21]. qRT-PCR analyses for the auxin response were performed to investigate the underlying molecular mechanisms, because flg22-mediated root growth inhibition was affected at the region of the elongation zone. In addition, previous studies also showed that flg22-induced miRNA393 leads to the posttranscriptional inhibition of the F-box auxin receptors, such as *TRANSPORT INHIBITOR RESPONSE1* (*TIR1*) and *AUXIN SIGNALING F-BOX PROTEIN2/3* (*AFB2/3*), required for plant growth via the auxin response [22]. Because MAMP signal, such as flg22, usually triggers immune responses with growth retardation via miRNA-mediated auxin response, these results suggest a trade-off for balancing between plant growth and immunity [23,24]. Consistent with previous results [22], the expression of auxin receptor gene *AFB3* and auxin-responsive gene *GRETCHEN HAGEN3.10* (*GH3.10*) was negatively regulated by the treatment of flg22 in WT compared to a control condition without flg22 but not *fls2 (*Figure 5A,B), suggesting that the flg22-mediated repression of auxin responses probably led to a decrease in root growth for optimizing plant fitness. In addition, because other G protein mutants also displayed similar decreased patterns of gene expression compared to each control condition, these results indicate that the defects of pathogen-mediated root growth related to the auxin response are not mediated by the role of heterotrimeric G proteins.

*PHYTOCHROME INTERACTING FACTOR4* (*PIF4*), which encodes a basic helix–loop–helix (bHLH) transcription factor (TF), is known to be regulated by growth-promoting hormones, such as brassinosteroid and gibberellin, for cell elongation [25,26]. In addition, *PIF4* functions as an upstream regulator of auxin biosynthetic genes, such as *YUCCA8* (*YUC8*), and regulates auxin signaling through *INDOLE-3-AETIC ACIDs* (*IAAs*) and *AUXIN RESPONSE FACTORs* (*ARFs*) [27,28,29]. Based on our findings (Figure 5A,B), which demonstrated that auxin is not correlated with flg22-mediated growth inhibition via G protein signaling, we examined *PIF4* expression as an upstream regulator of auxin. As expected, the flg22 treatment leading to reduced root growth repressed *PIF4* expression in WT but not *fls2* (Figure 5C). Although the decreased rate of *PIF4* expression in *gpa1-4* (60%) was similar to that in WT (62%), *PIF4* expression in *agb1-2* and *gpa1-4 agb1-2* exhibited less sensitive reduction rates, representing 53% and 49%, respectively, compared to each control condition (Figure 5C). These less sensitive expression patterns of *PIF4* in *agb1-2* and *gpa1-4 agb1-2* appear to coincide with the less sensitive defects in the length of the elongation zone and primary roots (Figure 1 and Figure 4). Therefore, although genetic evidence is not supported in this study, these results suggest that *PIF4* is partially involved in flg22-mediated growth inhibition via G protein signaling. Moreover, because we cannot rule out the redundant effect other than *PIF4*, the downstream genes involved in growth regulation via G protein signaling under biotic stress conditions will be further investigated. 

### 2.4. The Role of G Protein Signaling Is Involved in FCA-Mediated Thermomorphogenesis under Heat Stress Conditions 

In addition to biotic stress, such as flg22, the biological role of G protein signaling in abiotic stress-mediated growth inhibition was further investigated. The *Arabidopsis* RNA-binding protein FCA functions as a critical activator in the autonomous flowering pathway by repressing a major floral repressor *FLOWERING LOCUS C* (*FLC*) [30]. A recent report showed that *FCA* plays a vital role in the thermal adaptation of plant growth and development under mild heat stress conditions by regulating oxidoreductases to reduce reactive oxygen species (ROS) [31]. In this study, the *fca-9* mutant exhibited retarded seedling growth at 28 °C [31]. Therefore, to determine if growth regulation mediated by heterotrimeric G proteins is also applied to *FCA*-mediated thermal adaptation, we carried out double and triple mutant analyses with *gpa1-4 fca-9*, *agb1-2 fca-9,* and *gpa1-4 agb1-2 fca-9* established by genetic crosses. Consistent with previous results [31], seedling growth of *fca-9* was severely retarded at 28 °C (Figure 6A and Appendix A). In contrast, heterotrimeric G protein mutants, such as *gpa1-4*, *agb1-2*, and *gpa1-4 agb1-2*, were indistinguishable from WT, including primary root growth under 22 °C and 28 °C conditions (Appendix A). These results suggest that G protein signaling is unrelated to general thermomorphogenesis. On the other hand, the growth defect of *fca-9*, including primary root growth, under heat stress conditions was recovered in *gpa1-4 fca-9* but not *agb1-2 fca-9* and *gpa1-4 agb1-2 fca-9* (Figure 6A). Moreover, in contrast to the response on flg22-mediated root growth inhibition, the reduced root meristem activity by showing the decreased division zone under heat stress conditions was found in *fca-9*, and this defective lesion was rescued only in *gpa1-4 fca-9* (Figure 6B). Therefore, these results suggest that *FCA*-mediated thermomorphogenesis under heat stress conditions depends on the role of *GPA1*. Furthermore, although the genetic evidence is not complete, it is likely that *agb1-2* is epistatic to *gpa1-4* in the *fca-9* background, as the rescued phenotype of *gpa1-4 fca-9* disappears in *gpa1-4 agb1-2fca-9*. 

## 3. Discussion

### 3.1. The Diverse Roles of G Protein Signaling in Plant Immunity 

Heterotrimeric G proteins as signaling elements have been known to function in the defense responses [10]. Previous studies showed that two *Arabidopsis* G proteins, such as GPA1 (Gα) and AGB1 (Gβ), have an opposite effect on the defense responses against necrotrophic fungal pathogens, including *Plectosphaerella cucumerina*, *Botrytis cinerea*, and *Fusarium oxysporum* [32,33]. For example, the *gpa1* mutants showed slightly increased resistance, whereas the *agb1* mutant exhibited increased susceptibility to fungal pathogens [32,33]. The loss-of-function double mutant of *AGG1* and *AGG2*, encoding a Gγ subunit, also showed similar susceptibility to the *agb1* single mutant to *P. cucumerian*, *F. oxysporum,* and *Alternaria brassicicola* [34,35], suggesting that the Gβγ dimer contributes more significantly to the defense responses against the fungal pathogens in *Arabidopsis*. In contrast to the fungal pathogens, the *gpa1* and *agb1* mutants were similar to WT after inoculation of *Pseudomonas syringae* pv *tomato* strains DC3000 [33], suggesting that the defense signaling mediated by heterotrimeric G proteins is independent of hemibiotrophic bacterium *Pst* DC3000. Recently, protease IV (PrpL), secreted from the opportunistic bacterial pathogen *P. aeruginosa*, activated a novel immune pathway, including MAPK activation and gene expression [36]. In this pathway, heterotrimeric G protein complexes function as an upstream of an MAPK cascade through interactions between the Gβ subunit and the receptor for activated C kinase 1 (RACK1), acting as a novel scaffolding protein for all three layers of MAPK cascade enzymes [36,37]. Interestingly, the PrpL-G protein-triggered signaling module is distinct from previously established immune signaling pathways elicited by MAMP signal flg22 because of the role of the heterotrimeric G proteins. Although the PrpL-triggered MAPK activation was abolished in *gpa1-4 agb1-2* double mutants, the immune responses, including MAPK activation and downstream immune gene expression, induced by flg22 were indistinguishable between WT and G protein mutants in previous studies (Figure 2) [36], suggesting the complicated and diverse roles of G protein signaling in plant innate immunity. 

MAMP-triggered immunity induces downstream MAPK activation and immune gene expression, followed by growth inhibition for the trade-off between growth and biotic/abiotic stress response because of limited resources [23,24,38,39]. Consistent with previous results shown by *Pst*DC3000, this study revealed a similar induction of MAPK activation and the expression of immune genes, such as *WRKY30* and *MYB51*, in WT and G protein mutants, including *gpa1-4*, *agb1-2*, and *gpa1-4 agb1-2*, indicating less implication of G protein signaling in the flg22-mediated immune responses (Figure 2). On the other hand, flg22-mediated growth inhibition was partially attenuated in G protein mutants (Figure 1). In addition, this growth inhibition in roots was affected at the elongation zone rather than the division zone (Figure 3 and Figure 4). Whereas the division zone related to cell division was indistinguishable between WT and G protein mutants, the length of the elongation zone by flg22 reflected the overall growth of primary roots. An elongation zone is important for exuding secondary metabolites acting as attractants or antimicrobials for microbes [40]. A previous study reported that the elongation zone acts as a place for tissue-specific MAMP responses in roots, including the inducible expression of immune genes, such as *CYP71A12* and *MYB51* [41], indicating similar MAMP responses by flg22 in roots via the elongation zone. Therefore, these results suggest that G protein signaling may be involved in the indirect MAMP-triggered growth inhibition to divert limited energy to the stress response rather than direct immune response after perceiving MAMP signals. 

### 3.2. The Diverse Roles of G Protein Signaling in Plant Growth 

Besides biotic stresses, growth phenotypes associated with abiotic stresses in G protein mutants have also been found in previous studies. Under ozone stress conditions inducing oxidative burst in *Arabidopsis*, *gpa1-4* and *agb1-2* mutants showed oppositely less and more sensitive tissue damage on their growth than WT, respectively [42]. Oxidative burst elicited by ozone treatment appeared as two peaks in *Arabidopsis* WT over time, as measured by the production of ROS, e.g., hydrogen peroxide. Interestingly, the first peak was diminished in both *gpa1-4* and *agb1-2* mutants. In contrast, the second peak declined in *gpa1-4* but not *agb1-2*. These results suggest that the first oxidative burst is required for the role of heterotrimeric G proteins, while the second oxidative burst is predominantly caused by the Gα subunit. Therefore, these results revealed the more resistant growth of *gpa1-4* under ozone stress conditions. Meanwhile, a recent study also showed that the function of the G protein β subunit, *AGB1*, is negatively involved in drought tolerance by downregulating the expression of ABA-responsive genes, including *MPK6*, *VIP1*, and *MYB44* [17]. In this study, *agb1-2* showed enhanced drought tolerance by displaying an increased survival rate and proline content compared to WT under drought stress conditions. In addition, the expression of ABA-responsive and drought-inducible genes were upregulated by the ABA and drought treatment in *agb1-2*. Overall, these results suggest that G protein signaling plays a role in the abiotic stress responses and tolerance related to plant growth in various ways. 

Basically, G protein signaling in plants has been known to function in developmental growth, morphology, and responses to hormones and stresses [10,43]. In previous studies, *Arabidopsis gpa1-4* and *agb1-2* mutants displayed developmental defects, such as hypocotyl length and hook opening in skotomorphogenesis and seed germination, rosette leaf size/shape, flower, silique length, lateral root formation, and entire root mass under normal growth conditions [8,44,45,46,47], suggesting the various roles of G protein signaling in fundamental plant growth and development. Although the Gα (GPA1) subunit and Gβ (AGB1) subunit form a heterotrimeric G protein complex, the functional importance of *GPA1* and *AGB1* involved in the stress responses associated with plant growth has been found to be either or both of them by mutant analyses [17,32,33,42]. Consistently, the lateral organ number of *gpa1* and *agb1* mutants decreased and increased, respectively, under normal growth conditions [47]. In addition, the *gpa1 agb1* double mutant showed increased lateral roots similar to the *agb1* single mutant [47], showing that *agb1* is epistatic to *gpa1* with an opposite effect on lateral organ formation. Based on these studies, our results suggest not only that *AGB1* is more significantly involved in growth inhibition triggered by MAMPs, but also that *GPA1* and *AGB1* have a genetic relationship through showing the same phenotype between *gpa1-4 agb1-2* and *agb1-2* by the flg22 treatment (Figure 1). Moreover, both *GPA1* and *AGB1* function in *FCA*-mediated growth inhibition under heat stress (Figure 6), although the loss of function of *AGB1* in the *fca-9* background was indistinguishable from *fca-9*. It is possible that the reduced effect is already masked by an excessively severe defect in the *fca-9* background under heat stress conditions. Although the underlying molecular mechanism remains elusive, these genetic analyses support the epistatic relationship between *GPA1* and *AGB1* with the opposite regulation in *FCA*-mediated growth inhibition under heat stress conditions.

## 4. Materials and Methods 

### 4.1. Plant Materials and Growth Conditions

Columbia-0 (Col-0) is the wild-type (WT) *Arabidopsis* plant used in this study. Seeds of *gpa1-4*, *agb1-2*, and *gpa1-4 agb1-2* as well as *fls2* (Salk_141277) were obtained from the ABRC (Arabidopsis Biological Resource Center, Columbus, OH, USA). They are all in the Col-0 background. Double or triple mutants, including *gpa1-4 fca-9*, *agb1-2 fca-9*, and *gpa1-4 agb1-2 fca-9*, were prepared by genetic crosses and confirmed by PCR genotyping. Primer information for genotyping is shown in Appendix A. The *fca-9* allele as a cleaved amplified polymorphic sequence (CAPS) marker was confirmed after the cleavage with *Sty*I (New England Lab, Ipswich, MA, USA). The WT PCR products (383 bp) were cleaved into 374 bp and 9 bp, whereas the *fca-9* PCR products were separated into 299 bp, 75 bp, and 9 bp after *Sty*I digestion. For root growth assays and qRT-PCR under flg22 conditions, seeds were sowed and grown vertically on half MS plates (0.5× MS, 1% sucrose, and 0.3% Gelrite, pH 5.7 adjusted with KOH) without (control) or with 100 nM flg22 for 7 days at 22 ± 0.8 °C with 100 μmol m^−2^ s^−1^ light intensity under 16 h light/8 h dark photoperiod conditions. For root growth assays under heat stress conditions, seeds were sowed and grown vertically on half MS plates at 22 ± 0.8 °C with 100 μmol m^−2^ s^−1^ light intensity (control) or 28 ± 1 °C for 7 days with 60 μmol m^−2^ s^−1^ light intensity under 16 h light/8 h dark photoperiod conditions. 

### 4.2. Root Growth Assay

To investigate the primary root growth, the distance between root caps to boundary points connected to hypocotyls in 7-day-old seedling roots grown vertically on half MS plates was measured using the ImageJ 1.53K (https://imagej.nih.gov/ij/download.html, (accessed on 16 June 2021)) program. For the analysis of the elongation zone length, 7-day-old seedling roots grown vertically on half MS plates were imaged using an optical microscope (Leica DM750, Wetzlar, Germany) to measure the length from the root cap to the differentiated root hair that has grown at least 50 µm in length using the ImageJ program.

For 5-ethynyl-2’-deoxyuridine (EdU) assay (Invitrogen/Thermo Fisher Scientific, Waltham, MA, USA), 7-day-old seedlings grown vertically on half MS plates were transferred to 6-well plates containing 1 mL of liquid medium (0.5× MS and 0.5% sucrose, pH 5.7 adjusted with KOH) for 1 h at room temperature. Seedlings were incubated with 1× EdU (10 µM) in liquid MS medium for 2 h at room temperature. After EdU incubation, seedlings were transferred to the new 1.5 mL tubes. Then, seedlings were fixed in fresh 4% paraformaldehyde solution for 15 min and washed twice with 1× PBS. After treatment with 0.1% Triton X-100 in 1× PBS for 15 min, seedlings were washed twice with 1× PBS and treated with EdU reaction cocktails of approximately 500 µL for 30 min at dark. The reaction cocktails were prepared with the following composition: 1.6 µL EdU buffer additive, 14 µL EdU reaction buffer, 6.7 µL CuSO_4_, 0.07 µL Alexa Fluor azide, and 144 µL distilled water. After washing the seedlings twice with rinse buffer and 1× PBS, respectively, the roots were cut and observed. The EdU images were taken using a fluorescence microscope (Leica DFC3000 G, Wetzlar, Germany), and the lengths of the EdU-labeled division zones were measured using the ImageJ program. 

For Lugol’s staining, roots of 7-day-old seedlings grown vertically on half MS plates were immersed in Lugol’s solution (Sigma-Aldrich, St. Louis, MO, USA) for 3 min. After staining, seedlings were washed with distilled water and roots were cut with a blade. Stained images were analyzed using an optical microscope (Leica DM750, Wetzlar, Germany).

### 4.3. Protein Extraction and Immunoblot Assay

Three or four seedlings frozen by liquid nitrogen were ground using a pellet pestle and cordless motor. Then, 100 μL of E buffer (135 mM Tris-HCl, pH 8.8; 1% (*w*/*v*) SDS; 10% (*v*/*v*) glycerol; 50 mM Na_2_S_2_O_5_) were added to extract total proteins. Samples were incubated on ice for 10 min and centrifuged at maximum speed for 10 min at 4 °C. The supernatant was transferred to a new 1.5 mL tube and 1/10 vol. of Z buffer (125 mM Tris-HCl, pH 6.8; 12% (*w*/*v*) SDS; 10% (*v*/*v*) glycerol; 22% (*v*/*v*) β-mercaptoethanol; 0.001% (*w*/*v*) bromophenol blue) was added. After performing SDS-PAGE, immunoblot assays were performed using an α-phospho-MAPK antibody (Cell signaling, MA, USA; Cat. No. #9101). Protein quantities were compared using an α-TUB antibody (Sigma-Aldrich, St. Louis, MO, USA; Cat. No. T9026).

### 4.4. Gene Expression Assay

For gene expression assays, 7-day-old seedlings grown on half MS plates were harvested and immediately frozen in liquid nitrogen. For *WRKY30* and *MYB51* expression, seedlings were transferred to 6-well plates containing liquid MS medium without or with 100 nM flg22. Seedlings were incubated for 1 h and then harvested. Total RNA were isolated from seedlings using TRI reagent (Molecular Research Center, OH, USA) according to the manufacturer’s instructions. First-stranded cDNAs were synthesized from 1 µg of total RNA using ImProm-II^TM^ reverse transcriptase (Promega, Madison, WI, USA). Primer information for the quantitative reverse transcription-polymerization chain reaction (qRT-PCR) assays is provided in Appendix A. The *ACT2* (*At3g18780*) gene was used as a control to normalize gene expression. qPCR reaction was performed on an MIC qPCR cycler (Bio Molecular Systems, Upper Coomera, QLD, Australia) using SYBR Green Realtime PCR Master Mix (TOYOBO, Osaka, Japan).

### 4.5. Statistical Analyses

All results requiring statistical analysis were analyzed by one-way analysis of variation (ANOVA) using SPSS (version 27) or Prism (version 7.0b) software. Significant differences and different letters among results were determined by one-way ANOVA with post hoc Tukey’s HSD or Duncan multiple comparison test.

## 5. Conclusions

This paper reported the physiological relevance of G protein signaling related to stress-mediated growth inhibition using heterotrimeric G protein mutants (Appendix A). Interestingly, the biological roles of heterotrimeric G proteins function differently under biotic and abiotic stress conditions. This complicated behavior of G protein signaling in plants may be caused by the different regulatory systems distinct from animal G protein signaling, even though both plants and animals possess the same G protein elements in the heterotrimeric complex. For this reason, downstream effectors of active Gα or Gβγ dimer are also not conserved in plants. Therefore, the effectors influenced by G protein subunits need to be further investigated to understand why plant G protein signaling is required for growth fitness under stress conditions. In addition, it will be possible to identify the common regulatory genes regulating stress-mediated growth by studying downstream genes modulated by heterotrimeric G protein subunits under biotic and abiotic stress conditions based on our results. Finally, this study provides new insights into G protein signaling related to the physiological trade-off between plant growth and stress for developmental plasticity under unfavorable stress conditions. 

## Figures and Tables

**Figure 1 ijms-24-11027-f001:**
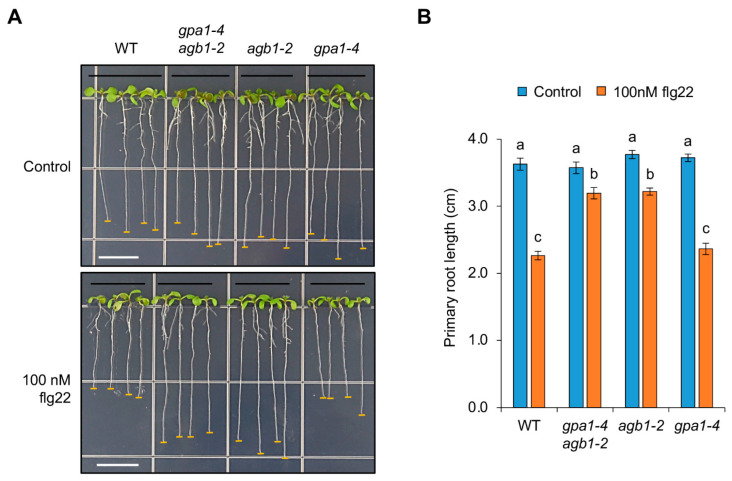
AGB1 is involved in the regulation associated with flg22-mediated root growth inhibition. (**A**) Root phenotypes of WT Col-0, *gpa1-4 agb1-2*, *agb1-2*, and *gpa1-4* seedlings grown vertically on half MS plates without (control) or with 100 nM flg22 for 7 days. White bars, 1 cm. (**B**) Primary root length of seedlings grown without or with 100 nM flg22. The data are presented as the mean ± S.E. (*n* = 14~20). The different letters indicate a significant difference (*p* < 0.05) according to the one-way ANOVA with post hoc Tukey’s HSD test.

**Figure 2 ijms-24-11027-f002:**
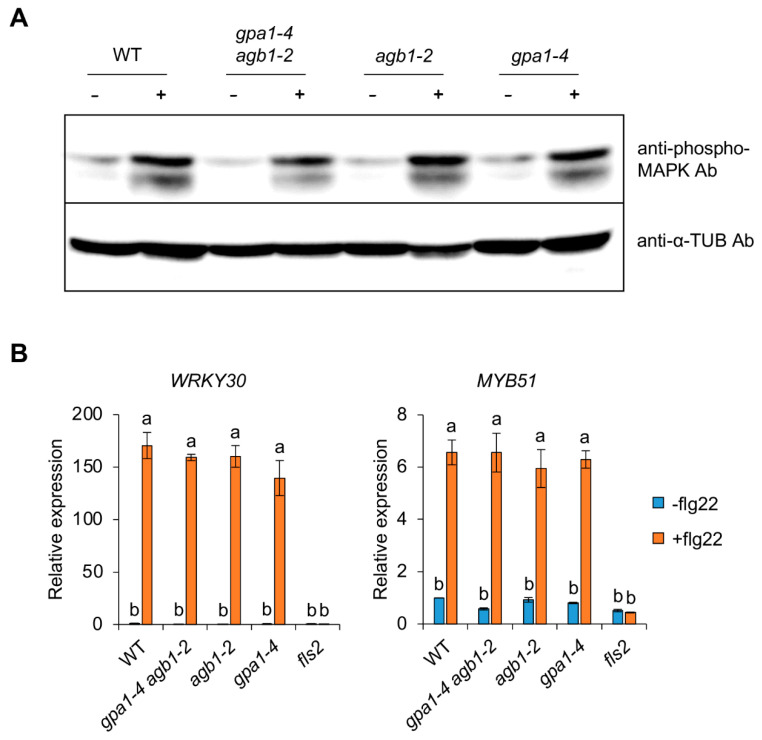
G protein signaling is unrelated to downstream responses after perception of flg22. (**A**) Immunoblot assays using WT, *gpa1-4 agb1-2*, *agb1-2*, and *gpa1-4* seedlings treated without (−) or with 100 nM flg22 for 10 min (+). Activated MAPKs were detected by an anti-phospho-MAPK antibody (Ab) and loading amounts of total proteins were analyzed by an anti-α-TUB antibody. (**B**) Expression of *WRKY30* and *MYB51* in WT, *gpa1-4 agb1-2*, *agb1-2*, *gpa1-4*, and *fls2* seedlings treated without (−flg22) or with 100 nM flg22 for 1 h (+flg22). The data are presented as the mean ± S.E. (*n* = 3~5). The different letters indicate a significant difference (*p* < 0.05) according to the one-way ANOVA with post hoc Tukey’s HSD test.

**Figure 3 ijms-24-11027-f003:**
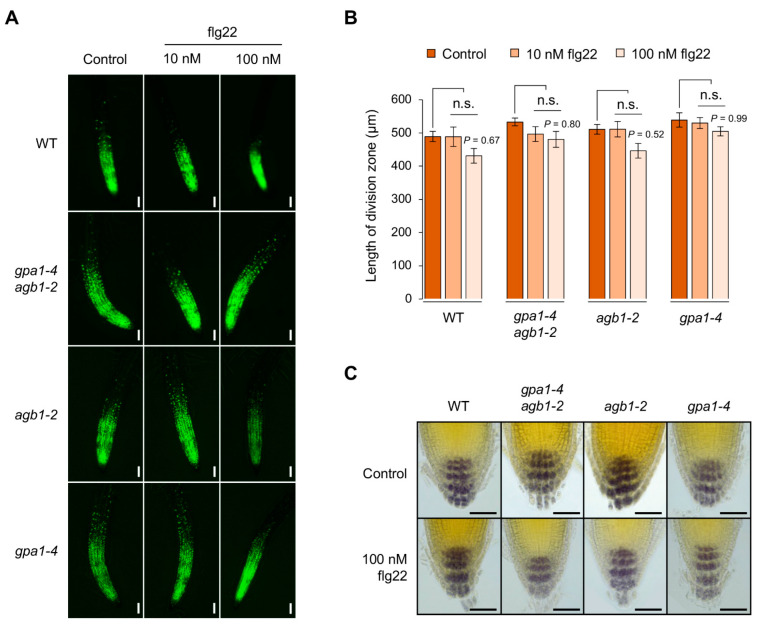
Cell division in root meristems is insensitive to flg22 treatment. (**A**) Root meristem phenotypes with proliferating cells labeled by EdU in WT, *gpa1-4 agb1-2*, *agb1-2*, and *gpa1-4* seedlings grown vertically without (control) or with 10 or 100 nM flg22 treatment for 7 days. Scale bars, 100 μm. (**B**) Quantified length of division zone in roots of seedlings grown without or with flg22 treatment. The data are presented as the mean ± S.E. (*n* = 6). Significance (*p* < 0.05) through multiple comparison was analyzed according to the one-way ANOVA with post hoc Tukey’s HSD test; n.s., not significant. (**C**) Lugol’s staining assay showing accumulation of starch granules in columella root cap of 7-day-old WT, *gpa1-4 agb1-2*, *agb1-2*, and *gpa1-4* seedlings treated without (control) or with 100 nM flg22. Scale bars, 50 μm.

**Figure 4 ijms-24-11027-f004:**
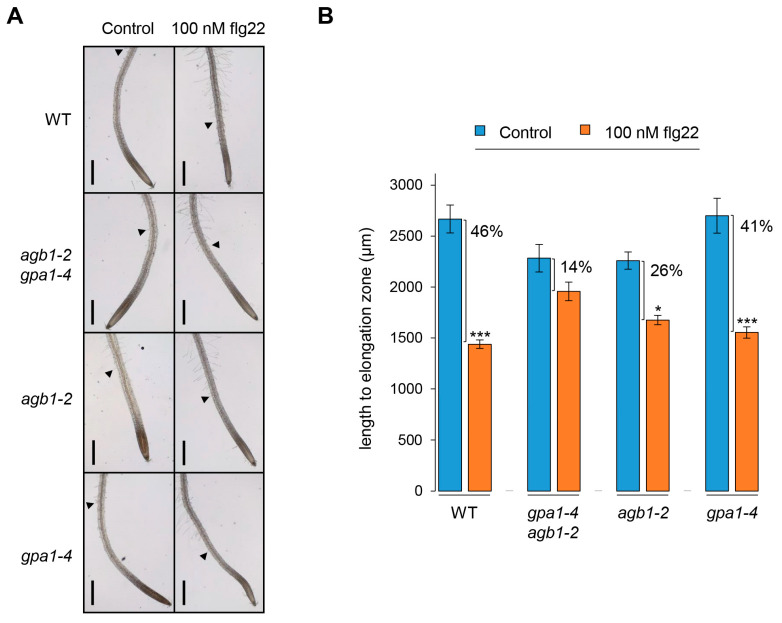
Elongation zone of roots is reduced by flg22 treatment. (**A**) Root phenotypes showing elongation zone of WT, *gpa1-4 agb1-2*, *agb1-2*, and *gpa1-4* seedlings grown vertically without (control) or with 100 nM flg22 for 7 days. Arrow heads indicate the root hairs that have grown at least 50 μm in length. Scale bars, 500 μm. (**B**) Quantified length of elongation zone in roots of seedlings grown without or with flg22. The data are presented as the mean ± S.E. (*n* = 8~18). Percentages indicate the reduced rate of elongation zone under 100 nM flg22 compared to each control condition. Significance through multiple comparison was analyzed according to the one-way ANOVA with post hoc Tukey’s HSD test. *, *p* < 0.05. ***, *p* < 0.001.

**Figure 5 ijms-24-11027-f005:**
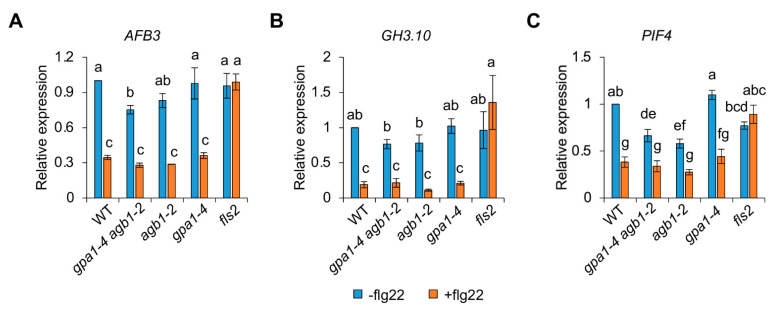
*AGB1* is partially involved in the repressed expression of *PIF4* by flg22 treatment. Expression patterns of *ABF3* (**A**), *GH3.10* (**B**), and *PIF4* (**C**) in WT, *gpa1-4 agb1-2*, *agb1-2*, *gpa1-4*, and *fls2* seedlings treated without (−flg22) or with 100 nM flg22 (+flg22) for 7 days. The data are presented as the mean ± S.E. (*n* = 3~6). The different letters indicate a significant difference (*p* < 0.05) according to the one-way ANOVA with post hoc Duncan test.

**Figure 6 ijms-24-11027-f006:**
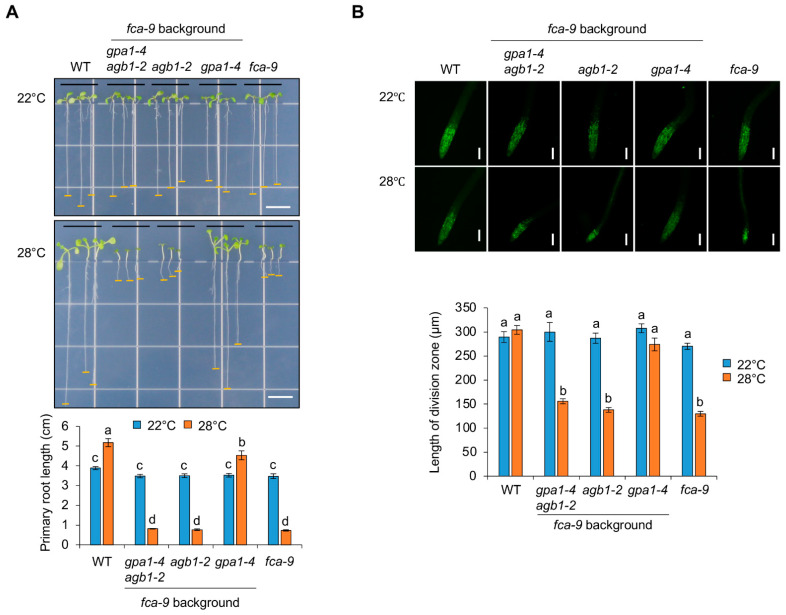
G protein signaling is related to FCA-mediated thermomorphogenesis. (**A**) **Top**, seedling phenotypes of WT, *gpa1-4 agb1-2 fca-9*, *agb1-2 fca-9*, *gpa1-4 fca-9*, and *fca-9* seedlings grown vertically on half MS plates under 22 °C or 28 °C conditions for 7 days. White scale bars, 1 cm. **Bottom**, primary root length of seedlings grown under 22 °C or 28 °C conditions. The data are presented as the mean ± S.E. (*n* = 10~12). (**B**) **Top**, root meristem phenotypes with proliferating cells labeled by EdU in WT, *gpa1-4 agb1-2 fca-9*, *agb1-2 fca-9*, *gpa1-4 fca-9*, and *fca-9* seedlings under 22 °C or 28 °C conditions for 7 days. Scale bars, 100 μm. **Bottom**, quantified length of division zone in roots of seedlings grown under 22 °C or 28 °C conditions. The data are presented as the mean ± S.E. (*n* = 9~18). The different letters indicate a significant difference (*p* < 0.05) according to the one-way ANOVA with post hoc Tukey’s HSD test.

## Data Availability

All data and materials presented in this paper are available on request from the corresponding author.

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
