# Peer review of "Heterotrimeric G Protein-Mediated Signaling Is Involved in Stress-Mediated Growth Inhibition in Arabidopsis thaliana"

_ijms, 2023, doi:10.3390/ijms241311027_

Round 1
Reviewer 1 Report
In this manuscript, authors demonstrated that AGB1 is invloved in flg22-mediated growth inhibition, and that FCA-mediated thermomorphogenesis under heat stress conditions depends on GPA1. The evidence supporting these two points seems to be convincing and solid. However, the point "Flg22-mediated inhibition of root growth through PIF4 is regulated in part by the role of G protein signalning" need more evidence to support, because no genetic data to show flg22-mediated inhibition of root growth through PIF4. It might be interesting to test flg22-mediated inhibition of root growth in pif4 mutant and PIF4 over-expression background.
It is interesting that fca-9 phenotype depends on GPA1 and agb1 mutation restore fca-9 gpa1 to fca-9. So I am wondering if AGB1 overexpression could restore fca-9 phenotype to WT?
In my view, there are some editorial errors in the manuscript. However, the way authors tried to describe their results are not so clear. For example, sentense in Page 9, line 256-258 is not accurate. The point of this sentense should be (1) FCA-mediated thermomorphogenesis under heat stress conditions depends on GPA1; (2) agb1-2 is epistatic to gpa1-4 in fca-9 background.
Author Response
Dear Editor and Reviewers,
We would like to submit our revised manuscript, entitled “Heterotrimeric G protein-mediated signaling is involved in stress-mediated growth inhibition in Arabidopsis thaliana” by Yang et al., for consideration of publication as an Article in International Journal of Molecular Science.
Most of all, we really appreciate the thoughtful comments and suggestions of the editor and reviewers. We have carefully considered them and followed their suggestions to address their concerns and improve our manuscript. Detailed responses to the reviewers' comments are provided with each comment. In addition, as suggested by the editor, we have also changed and deleted some sentences and words mainly in parts of Introduction and Materials and Methods to resolve the duplication issue.
To avoid confusion of corrections suggested by the editor or reviewers, we have used the “Track Changes” function with yellow highlighting for reviewers’ comments, whereas we have used only the “Track Changes” function without highlighting for editor’s comments. Also, as line numbering tends to be different between MS Word and PDF, we have included both cases in our responses.
Response to Reviewer 1 Comments
In this manuscript, authors demonstrated that AGB1 is invloved in flg22-mediated growth inhibition, and that FCA-mediated thermomorphogenesis under heat stress conditions depends on GPA1. The evidence supporting these two points seems to be convincing and solid. However, the point "Flg22-mediated inhibition of root growth through PIF4 is regulated in part by the role of G protein signalning" need more evidence to support, because no genetic data to show flg22-mediated inhibition of root growth through PIF4. It might be interesting to test flg22-mediated inhibition of root growth in pif4 mutant and PIF4 over-expression background.
Our response: We appreciate and agree with the reviewer's comments. As suggested by the reviewer, the role of PIF4 in flg22-mediated inhibition of root growth would require further genetic analysis. However, we do not currently have loss-of-function and gain-of-function mutants of PIF4. Furthermore, due to the limitations of the revision process (within 7 days), these analyses will be performed as future work. Instead, we have softened our claim. To do this, we have changed the subtitle of 2.3 (line 231 in Word; line 213 in PDF), the sentence (line 268-270 in Word; line 250-252 in PDF) and the conjunction (line 270 in Word; line 252 in PDF).
It is interesting that fca-9 phenotype depends on GPA1 and agb1 mutation restore fca-9 gpa1 to fca-9. So I am wondering if AGB1 overexpression could restore fca-9 phenotype to WT?
Our response: We appreciate the reviewer's comments. As suggested by the reviewer, we are also interested in whether overexpression of AGB1 in the gpa1-4 agb1-2 fca-9 triple mutant rescues the defective growth phenotype of fca-9 as shown in the gpa1-4 fca-9 double mutant (Figure 6A). If agb1-2 is epistatic to gpa1-4, AGB1 overexpression in gpa1-4 agb1-2 fca-9 could rescue the fca-9 heat stress phenotype as shown in gpa1-4 fca-9, such as a WT-like phenotype (Figure 6A), based on previous epistatic studies related to lateral root number (Chen et al., 2006; Reference #46). However, as we have no genetic evidence for this at present, we are currently preparing the AGB1 overexpression construct. Once we have the AGB1-OX construct, we will transform it into gpa1-4 agb1-2 fca-9.
In my view, there are some editorial errors in the manuscript. However, the way authors tried to describe their results are not so clear. For example, sentense in Page 9, line 256-258 is not accurate. The point of this sentense should be (1) FCA-mediated thermomorphogenesis under heat stress conditions depends on GPA1; (2) agb1-2 is epistatic to gpa1-4 in fca-9 background.
Our response: We appreciate and agree with the reviewer's comments. As suggested by the reviewer, we have changed and rewritten the sentence to make it clear (line 307-311 in Word; line 289-293 in PDF).
Response to Reviewer 2 Comments
Geldner lab loudly claimed that there no flg22 responses in roots, despite the literature showing otherwise and the present work adds important additional evidence that flg22 works on roots. The present work provides a mechanism through the G protein pathway.
Comments for improvement:
Line 28. Citing the original Hong Ma papers is kind but actually the original paper with the Southern blot turned out to be wrong data. Ma’s group should have picked up the XLG genes and so the conclusion should have been 4, not 1 G alpha gene. I would stick to the more recent reviews that cover this complexity in the plant G protein repertoire.
Our response: We appreciate the reviewer's comments. As suggested by the reviewer, we have added the word “canonical” for Gα protein (line 27 in Word; line 27 in PDF) and information of XLGs (line 27-28 in Word; line 27-28 in PDF) in the text. And we have changed reference #4 from Ma et al. to Ding et al. for XLGs. For the canonical Gα protein, we added the reference #1 (line 29 in Word; line 29 in PDF).
Line 50 D-glucose was never concluded to be a ligand to RGS1. The EC50 of 100 mM glucose is too large for glucose to have any signaling specificity. Rather, it is likely that some metabolite of glucose is the ligand if there is one at all.
Our response: We appreciate and agree with the reviewer's comments. Based on the reference, we have added the word “candidate” to restrict the meaning of ligand for D-glucose (line 86-87 in Word; line 68-69 in PDF).
Line 53. Reference 9 is a very strange review- it has a lot of speculation and I don’t think that is the best review to cite. Rather, because the sentence claims that all plant Galpha subunits have spontaneous exchange, it would be better to cite a paper that actually shows this. Urano’s lab showed this across the plant kingdom Bradford,W, Buckholz, A, Morton, J., Price, C, Jones, AM, Urano D. (2013) Ancestral regulation of eukaryotic G protein signaling. Science Signaling 6: ra37
Our response: We appreciate the reviewer's comments. As suggested by the reviewer, we changed reference #9 from Ofoe to Bradford et al.
Line 81: define FCA (Flowering Locus A)
Our response: We appreciate the reviewer's comments. At first, we have added the full name of FCA in section 2.4 of the submitted manuscript. Although I have been studying flowering time for quite a long time, I noticed that many publications mentioning FCA did not use the full name of FCA. As it was difficult to find the full name of FCA, we added the full name of FCA as FLOWERING CONTROL LOCUS A based on the description of TAIR web site (https://www.arabidopsis.org/servlets/TairObject?id=128853&type=locus). As suggested by the reviewer, although we found the full name of FCA in the previous publication as FLOWERING CA (Koornneef M, Current Biology 1997, 7:R651-R652), it did not seem to be in common use. Therefore, we think that the removal of the full name of FCA in the text should be fine. We have deleted the full name of FCA (line 279 in Word; line 261 in PDF).
Cite a supporting reference that AGB1 interacts with RACK such as . Klopffleisch, et al (2011) Molecular Systems Biology 7
Our response: We appreciate the reviewer's comments. As suggested by the reviewer, we have added the new reference for G-protein interaction with RACK as #37. Of course, the reference number has been renumbered accordingly after #37.
Line 367 Epistasis between fca-9 and agb1-2 is difficult to conclude but the discussion adequately points this out.
Our response: We appreciate the reviewer's comments. Although we have not changed the discussion section, we have changed and added the sentence regarding epistasis to make it clearer (line 308-311 in Word; line 290-293 in PDF).
Response to Reviewer 3 Comments
The manuscript under consideration is devoted to the analysis of the heterotrimeric subunits G-protein role in the signaling of plant responses to bioic and abiotic stress factors. Since the researchers had in their disposal a mutants collection of the model plants A.thaliana for the genes controlling the subunits of his protein, the authors had an excellent opportunity to plan experiment demonstrating their participation in the development of plant responses to flagellin, which models the responses to infection by some fungal pathogens. Of undoubted interest in this study are the results showing the participation of the G-protein in the regulation of plant growth processes under stress conditions, which opens up a new vector in the direction of work on the identification of downstream regulatory genes, the expression of which is modulated by subunits of the studied heterotrimeric G-protein. The experiments carried out are convincing, confirmed by good illustrations. The introduction presents the directions world community investigations on the topic of this manuscript, compares the role G-protein in animal and plants responses, and identifies similarities and differences in the mechanisms. As a wish, the authors of the manuscript could be recommended to summarize the conclusions of their experimental studies in the form of a graphical representation indicating the prospects for further direction on the study of the G-protein in plants.
Our response: We appreciate the reviewer's supportive comments. As suggested by the reviewer, we have added the graphical scheme in the Supplementary Material as Figure S2 and in the text (line 734 in Word; line 679 in PDF).

Reviewer 2 Report
Geldner lab loudly claimed that there no flg22 responses in roots, despite the literature showing otherwise and the present work adds important additional evidence that flg22 works on roots. The present work provides a mechanism through the G protein pathway.
Comments for improvement:
Line 28. Citing the original Hong Ma papers is kind but actually the original paper with the Southern blot turned out to be wrong data. Ma’s group should have picked up the XLG genes and so the conclusion should have been 4, not 1 G alpha gene. I would stick to the more recent reviews that cover this complexity in the plant G protein repertoire.
Line 50 D-glucose was never concluded to be a ligand to RGS1. The EC50 of 100 mM glucose is too large for glucose to have any signaling specificity. Rather, it is likely that some metabolite of glucose is the ligand if there is one at all.
Line 53. Reference 9 is a very strange review- it has a lot of speculation and I don’t think that is the best review to cite. Rather, because the sentence claims that all plant Galpha subunits have spontaneous exchange, it would be better to cite a paper that actually shows this. Urano’s lab showed this across the plant kingdom Bradford,W, Buckholz, A, Morton, J., Price, C, Jones, AM, Urano D. (2013) Ancestral regulation of eukaryotic G protein signaling. Science Signaling 6: ra37
Line 81: define FCA (Flowering Locus A)
Cite a supporting reference that AGB1 interacts with RACK such as . Klopffleisch, et al (2011) Molecular Systems Biology 7
Line 367 Epistasis between fca-9 and agb1-2 is difficult to conclude but the discussion adequately points this out.
Author Response

(The authors gave the same response as above.)

Reviewer 3 Report
The manuscript under consideration is devoted to the analysis of the heterotrimeric subunits G-protein role in the signaling of plant responses to bioic and abiotic stress factors. Since the researchers had in their disposal a mutants collection of the model plants A.thaliana for the genes controlling the subunits of his protein, the authors had an excellent opportunity to plan experiment demonstrating their participation in the development of plant responses to flagellin, which models the responses to infection by some fungal pathogens. Of undoubted interest in this study are the results showing the participation of the G-protein in the regulation of plant growth processes under stress conditions, which opens up a new vector in the direction of work on the identification of downstream regulatory genes, the expression of which is modulated by subunits of the studied heterotrimeric G-protein. The experiments carried out are convincing, confirmed by good illustrations. The introduction presents the directions world community investigations on the topic of this manuscript, compares the role G-protein in animal and plants responses, and identifies similarities and differences in the mechanisms. As a wish, the authors of the manuscript could be recommended to summarize the conclusions of their experimental studies in the form of a graphical representation indicating the prospects for further direction on the study of the G-protein in plants.
Author Response

(The authors gave the same response as above.)

Round 2
Reviewer 1 Report
The authors addressed all the comments and so this manuscript is suiable for IJMS in my view.